# Tb^3+^ Ion Optical and Magneto-Optical Properties in the Cubic Crystals KTb_3_F_10_

**DOI:** 10.3390/ma15227999

**Published:** 2022-11-12

**Authors:** Uygun V. Valiev, Denis N. Karimov, Chong-Geng Ma, Odiljon Z. Sultonov, Vasiliy O. Pelenovich

**Affiliations:** 1Faculty of Physics, National University of Uzbekistan (NUUz), Tashkent 100174, Uzbekistan; 2Shubnikov Institute of Crystallography, Federal Scientific Research Centre «Crystallography and Photonics», Russian Academy of Sciences, Moscow 119333, Russia; 3School of Optoelectronic Engineering & CQUPT-BUL Innovation Institute, Chongqing University of Posts and Telecommunications, Chongqing 400065, China; 4Institute of Technological Sciences, Wuhan University, Wuhan 430072, China

**Keywords:** KTb_3_F_10_ crystal, Tb^3+^ ion, crystal field, optical transitions, Stark sublevels, magnetic circular dichroism, absorption spectra, magnetic circularly polarized luminescence

## Abstract

The optical and magneto-optical characteristics of KTb_3_F_10_ crystals in the transition region of ^5^D_4_ → ^7^F_6_ 4*f*^8^ configurations of the Tb^3+^ ion at temperatures of 90 and 300 K were studied. The schemes of the optical transitions in the KTb_3_F_10_ crystals were constructed, and the energies of most of the Stark sublevels of the ground ^7^F_6_ and excited ^5^D_4_ multiplets of the Tb^3+^ ion split by the C_4v_ symmetry crystal environment were determined. The presence of three- and two-doublet states in the energy spectra of the Tb^3+^ion multiplets ^7^F_6_ and ^5^D_4_, respectively, was established, which is in good agreement with theoretical predictions. The use of the wavefunctions of the Stark sublevels of multiplets split by a tetragonal crystal field and combining in the studied optical transition made it possible to explain some of the magnetic and magneto-optical features observed in the KTb_3_F_10_ single crystals.

## 1. Introduction

The development of new paramagnetic crystals based on rare-earth ions, which have high optical transparency and efficient rotation of the polarization plane under the magnetic field action, is still relevant [1]. Terbium-containing concentrated fluoride crystals, due to their better thermo-optical characteristics in comparison with oxides (primarily terbium-gallium garnet crystals [2,3]), are very promising materials for creating effective optical Faraday insulators based on them in high-power laser systems and currently represent a real alternative to the Tb_3_Ga_5_O_12_ crystal for practical application in the functional magneto-optical devices. KTb_3_F_10_ (KTF) crystals [4,5,6,7,8] are optically isotropic (space group *Fm*–3*m*), and this distinguishes them favorably from other actively studied anisotropic fluorides based on both Tb^3+^ ions (such as TbF_3_ [9,10,11,12], LiTbF_4_ [13,14]) and other rare earth element (REE) ions (CeF_3_, PrF_3_) [14,15,16,17]. In addition, these crystals are very interesting for their use as active and passive elements in photonics and X-ray scintillators [18,19,20] because, according to the data of [19], KTb_3_F_10_ crystals are characterized by bright “yellow-green” luminescence, easily excited by both arc (Hg and Xe) lamps and X-ray radiation.

Bulk KTF single crystals are successfully grown by various directional crystallization methods [3,21,22]. The main difficulties associated with the incongruent melting of the compound and the tendency to pyrohydrolysis of the initial fluoride components have been successfully overcome to solve the problem of obtaining KTF crystals of the required optical quality. Now the industrial production of these crystals by the modified Czochralski method is carried out by Northrop Grumman SYNOPTICS (USA) [3]. The use of KTF crystals in practical applications requires a detailed study of their optical and magneto-optical properties, which will contribute to a deep understanding of the fundamental features of the magneto-optics of REE ions in crystalline fluoride materials.

Therefore, the motivating factor of this study was both the growth of bulk KTF optical quality crystals from a melt, and a detailed analysis of the experimental data of optical (absorption, luminescence) and differential methods of magneto-optical studies, including magnetic circular dichroism—MCD and magnetic circular polarization of luminescence—MCPL, carried out in the visible range of the spectrum on the obtained samples of double potassium-terbium fluoride crystals. In our opinion, the use of magneto-optical research methods has significantly expanded the capabilities of traditional optical methods previously used by Pues et al. [19] to the REE compound under consideration, which made it possible to establish both the presence of doubly degenerate states (doublets) and their energies in the spectra of ^5^D_4_ and ^7^F_6_ multiplets of Tb^3+^ ions localized in the tetragonal crystal field of C_4v_ symmetry of KTF crystals. It is this experimental discovery of doublet states in the spectra of ^5^D_4_ and ^7^F_6_ multiplets that made it possible to give a consistent explanation for the appearance of features in the magnetic and magneto-optical properties of KTF crystals.

## 2. Materials and Methods

### 2.1. Crystal Growth and Their Characterization

KTF crystals were grown by vertical directional crystallization in a double-zone resistive equipment with a graphite heating set in a mixed fluorinating atmosphere. A detailed description of the growth equipment is given in [22]. Anhydrous powder TbF_3_ (99.99%, Lanhit Ltd., Moscow, Russia) and KF (≥99.9%, Merck KGaA, Darmstadt, Germany) were utilized as initial reagents. The powders were pre-annealed in a vacuum and melted in a fluorinating (He + CF_4_ + HF) atmosphere for purification from oxygen-containing impurities. Growth was carried out from a melt enriched with KF (charge composition about 28/72 mol %). The pulling rate of the crucible was 3 mm/h, and evaporation losses were less than 1 wt. %.

X-ray diffraction (XRD) analysis of crystals was carried out on a powder X-ray diffractometer MiniFlex 600 (Rigaku, Tokyo, Japan) with CuKα radiation. Diffraction peaks were recorded in the angular range 2θ from 10 to 100°. Crystal phases were identified using ICDD PDF-2 (2014). Le Bail full-profile analysis method was applied to calculate the crystal unit-cell parameters (Jana2006 program (2014), the *Fm*–3*m* space symmetry group).

As a result of growth experiments, transparent KTF crystals with a length of 30–50 mm and a diameter of up to 30 mm were obtained, which do not contain any cracks and scattering inclusions (Figure 1, insert). The grown crystals were single-phase, and the cubic lattice parameter was a = 11.6732(2) Å at room temperature (Figure 1). The samples for the study were polished plates 0.5–5 mm thick and oriented along the cleavage plane (111).

Crystal transmission spectra were recorded at room temperature (RT) using a Cary 5000 UV-Vis-NIR spectrophotometer (Agilent Technologies, Inc., Santa Clara, CA, USA) in the wavelength range λ = 0.19–3.00 µm (52,640–3333 cm^−1^). Spectral features characteristic of the Tb^3+^-containing crystalline materials was observed [6,9]. The KTF crystal is characterized by a wide operating transparency range in the range λ = 0.4–1.6 µm (Figure 2), with the exception of the absorption band (transition ^7^F_6_ → ^5^D_4_) in the region λ~0.485 µm (20,620 cm^−1^). Absorption in the IR range of the spectrum is associated with transitions from the ground ^7^F_6_ state to the ^7^F_j_ (j = 0–6) 4*f*^8^ levels of the Tb^3+^ ion configuration. High-lying transitions within the 4*f*^8^ configuration, located in the UV spectral region, were observed, which overlap with intense bands determined by transitions to the lower levels of the 4*f*^7^5*d* configuration of the Tb^3+^ ion.

### 2.2. Optical and Magneto-Optical Investigation Methods

Optical (absorption, photoluminescence (PL)), and magneto-optical (magnetic circular dichroism (MCD), magnetic circular polarization of luminescence (MCPL)) spectra for KTF crystals were measured at temperatures *T* = 90 and 300 K. In order to better resolution of the fine structure of the spectra, a diffraction monochromator MDR–23 (LOMO, St. Petersburg, Russia) was used. The spectral characteristics were precisely measured in the absorption (or emission) band attributed to the 4*f* transition ^5^D_4_ → ^7^F_6_ (forbidden in the so-called “electric-dipole approximation”). The range from 480 to 500 nm (20,833–20,000 cm^−1^) with an average instrument spectral resolution of 0.05 nm (2 cm^−1^) for PL and absorption measurements was covered by optical low-temperature experiments. The relative error of the spectral registration was about 2–3%.

Ultraviolet (UV) radiation emitted by a DRL Hg-lamp (power 250 W) and limited using a UFS-5 optical filter in the range 240–400 nm (41,650–25,000 cm^−1^) was applied to visible PL excitation of KTF crystal, which located predominantly in the blue (transition ^5^D_4_ → ^7^F_6_) and green (transition ^5^D_4_ → ^7^F_5_) spectral regions (Figure 3). Terbium-containing fluoride is an efficient converter of UV and X-ray [19] radiation into visible light.

The so-called natural (completely unpolarized) light radiation is used in magneto-optical measurements of passing (MCD) or emitted (MCPL) light. The orientation of the magnetic field *H* is parallel to the light propagation. Light passing from a crystal in a magnetic field becomes partially polarized. The degree of circular polarization was measured using a photoelastic modulator with the resonance frequency of Ω = 36 kHz and a linear polarizer, as described earlier in Ref. [23]. The voltage modulator supply was chosen in the range of 30–90 V and determined by the wavelength of the light.

### 2.3. Experimental Setups

As a light source to measure MCPL spectra 250 W Hg lamp was utilized. The light radiation was collected by the quartz condenser mounted near the lamp. Next, the light radiation was passed through a water filter to exclude the IR radiation. After passing UV filters (bandpass of 250–400 nm), the radiation was focused on the sample, which was fixed in the air gap of the electromagnet, and finally excited the PL. In the presence of the magnetic field, the PL of the sample became partially polarized. After the sample, an objective lens was used to collect the partially polarized PL light. Next, this light passed through the PE modulator. A linear polarizer after the modulator was used as an analyzer. Finally, the modulated light was focused on the input slit of the monochromator MDR–23. The analyzer transmission plane was rotated 45° relative to the induced optical axis of the PE modulator. The photomultiplier (PMT) was operated with a constant average anode current. For such an operating mode, we used an adjustable voltage power supply, which supplied the PMT anode circuit with negative feedback. The power supply provided stabilization of the anode current at a level of 1–2%. In such a constant current mode, the measured ratio of *U*(Ω)/*U*(0) becomes proportional to the *U*(Ω) since the value of *U*(0) is constant regardless of the level of illumination of the PMT photocathode [24]. For measurement of the MCD spectra in the visible region, we used a halogen lamp, and other experimental devices were completely the same.

The same experimental setup was also used for the measurement of the absorption spectra. The registration of absorption spectra in the visible region was carried out using the PMT sensitized using the previously described technique, which stabilizes the average PMT current when scanning along a given line shape [11,23]. In realization of the PMT current stabilization ($i_{p}=const$) due to the feedback between the *dc* amplifier connected with the output of the PMT and its high voltage source, one can obtain a ratio between the relative change ΔI/I of light intensity *I* passing the sample and PMT voltage supply *U* of the PMT.

The PMT anode current as a function of the light intensity *I* and power supply voltage *U*, in the region of linear current-illumination dependency, can be written as:(1)$$i_{p}=AI\varphi (U)$$
where *A* is a scale factor, and φ(U) can be written as $\varphi (U)=B\cdot 10^{kU}$, where *B* and *k* are scale factors. The φ(U) is one of the main characteristics of the PMT, which characterizes anode sensitivity. Since the averaged anode current $i_{p}=const$, we can obtain the ratio between the change in the PMT voltage supply ΔU and the optical density of sample *D*: kΔU=ΔI/I=KD, where *K* is a calibration coefficient [23,24]. Therefore, in the PMT current stabilization mode, a change in the PMT voltage ΔU describes the change in the illumination ΔI/I, which is proportional to the optical density of the sample. The coefficient *K* should be found in the calibration procedure of the spectrophotometer by using optical filters with known absorption coefficients.

An *e*-shape sectional core electromagnet was used to obtain the magnetic field [25]. By using this magnet, experiments in longitudinal (the light wave vector is a parallel magnetic field) or transverse (the light wave vector is perpendicular) geometry can be carried out. Moreover, transmission or reflection modes are also available. The maximal magnetic field is 1.2 T, and both polarities are available; the air gap is 6 mm. The magnet has an expandable air gap to mount a cryostat. The custom build cryostat is made of thin-walled stainless steel and filled with liquid nitrogen necessary for the measurements of the MCD and MCPL temperature dependences [26]. This cryostat was fixed in the air gap of the electromagnet. The maximal magnetic field with the cryostat is 0.5 T. The InSb–based Hall sensor (operating area 0.45 × 0.15 mm^2^, and sensitivity 114 µV/mT) was used to control the magnetic field.

## 3. Experimental Results and Discussion

### 3.1. Absorption and PL Spectra of KTF Crystal in the Visible Spectral Region

The optical data indicate that all detected PL bands under UV excitation in the crystal (at *T* = 90 K) are almost completely concentrated in the wavelength range of 482–494 nm (20,750–20,240 cm^−1^), determined by the ^5^D_4_ → ^7^F_6_ transition in the Tb^3+^ ions (Figure 4). No additional spectral features were observed in the short wavelength region.

The KTF compound crystallizes in a fluorite-type structure (space group *Fm*–3*m*, Z = 8). The asymmetric unit cell contains one rare-earth Tb^3+^ cation, one K^+^ anion, and two different fluorine ions [19,22,27,28]. The crystal structure can be described by polyanion clusters of six square TbF_8_ antiprisms connected either by edges—a cluster [Tb_6_F_32_]—or by a three-dimensional assembly of clusters [Tb_6_F_36_] and a polyhedron KF_16_. Rare-earth Tb^3+^ cations in the structure are bound to eight fluorine ions, forming square antiprisms, which leads to the appearance of equivalent tetragonal optical centers characterized by a point symmetry group C_4v_ [29].

Due to the effect of the ligand (crystal) field of such symmetry, the degeneracy of various energy states of Tb^3+^ ions doped into the KTF matrix is not completely removed, but only partially. Indeed, by using group-theoretic considerations [29], it can be shown that the lowest multiplet ^7^F_6_ of the basic configuration 4*f*^8^ of the Tb^3+^ ion splits into 10 Stark sublevels: 2A_1_ + A_2_ + 2B_1_ + 2B_2_ + 3E. Sublevels A and B are singlet states, and the irreducible representations of A and B differ from each other in that the characters of the first do not change the sign when rotating around the main axis of symmetry (C_4_ in our case), and the second change the sign. In this case, irreducible representations of E characterize the doublet states. Note that A_1_ and A_2_ representations behave differently during reflection operations in the symmetry plane (the same as B_1_ and B_2_). At the same time, the excited state of ^5^D_4_ belonging to the same electronic configuration is split into seven sublevels: 2A_1_ + A_2_ + B_1_ + B_2_ + 2E. If doublet is the ground state of the REE ion, the magnetization of the rare-earth *M_H_* subsystem (per one ion) in the crystal at low temperatures in the direction of the magnetic field *H* parallel to the z-axis (C_4_) can be described by the expression [30,31]:(2)$$M_{H}=\frac{1}{2}g\mu_{B}th\left(\frac{g\mu_{B}H}{2k_{B}T}\right)$$
where the coefficient *g* is called the g-factor, or, in relation to a free atom, the Lande factor; $\mu_{B}$ is Bohr magneton; and $k_{B}$ is Boltzmann constant.

A detailed examination of the data of magnetic measurements of the KTF crystal at low temperatures [19] shows that Formula (2) qualitatively correctly describes the behavior of the field dependencies of the magnetization at temperatures in the range of 3–50 K, which indicates the presence of the lowest doublet state in the ^7^F_6_ multiplet spectrum. The intense absorption lines shown in Figure 4, indicated by marks with numbers 1–4 and 6, correspond to transitions from the ground state of the ^7^F_6_ multiplet to the Stark levels of the excited ^5^D_4_ multiplet split by the crystal field. Lines 5 and 7 arise during the transition from levels with an energy of 22 and 82 cm^−1^, respectively. The weak absorption lines (lines 8 and 9) are caused by the optical transitions from the excited sublevels of the ^7^F_6_ multiplet.

The superposition of the measured PL and absorption spectra associated with optical transitions between the ^5^D_4_ and ^7^F_6_ multiplets of the Tb^3+^ ion at *T* = 90 K is also shown in Figure 4. In these spectra, groups (or rather bands) of overlapping lines of different intensities and similar shapes are observed simultaneously for λ ≈ 485 and 486.5 nm (20,624 and 20,560 cm^−1^, respectively). It should be noted that the wavelengths of overlapping lines 3’, 4’, 5’, 6’, 7’, 8’ and 9’ observed in the PL spectra coincide with the wavelengths of lines 3, 4, 5, 6, 7, 8 and 9, found, respectively, in the absorption spectra. The above-mentioned coincidence of the energies of some lines in the studied optical spectra and the behavior of their temperature dependences facilitates the construction of a scheme of optical transitions in the absorption and PL spectra in the studied crystal (see Figure 5) and allows us to determine the energies of most of the Stark sublevels of the ground ^7^F_6_ and excited ^5^D_4_ multiplets of the Tb^3+^ ion split by the crystal symmetry environment C_4v_ (see Table 1). The transition from the free terbium atom to the Tb^3+^ ions located in the local C_4v_ symmetry positions in the KTF crystal leads to the appearance of Stark singlets (A_1_, A_2_, B_1_, B_2_) and doublets (E) in the energy spectra of the ground ^7^F_6_ and excited ^5^D_4_ multiplets of the Tb^3+^ ion. Further identification of optical transitions in KTF can be successfully carried out by placing the crystal under study in an external magnetic field *H*, which removes the twofold degeneracy of the doublet states E.

### 3.2. The MCD and MCPL Spectra of the KTF Crystal

It is well known that the perturbation caused by the external magnetic field manifests itself in the difference in the optical properties of the crystal for right- and left-circularly polarized light radiation. In this case, the difference in the corresponding refractive coefficients results in the Faraday effect (FE), which is the magneto-optical rotation of the light polarization plane, whereas the difference in absorption coefficients results in magnetic circular dichroism (MCD). In particular, according to [31,32], in the most general form, the MCD is defined by the expression:(3)$$\Delta \alpha =\left(\alpha_{+}-\alpha_{-}\right)=\gamma \mu_{B}H\left[\frac{A}{\hbar }\cdot \frac{df}{d\omega }+\left(B+\frac{C}{kT}\right)f\right]$$
where *α_+_* and *α*_−_ are the absorption coefficients of right− and left−circularly polarized light, respectively; γ is a constant [30,33]; *μ_B_* is the Bohr magneton; *ω* is the light frequency; *A, B* and *C* are the so-called MCD (or Faraday) parameters [33,34]; *f = f (ν,ν*_0_*)* is the function of the absorption band contour; *k_B_* is the Boltzmann constant.

The occurrence of the *A*—term in Formula (3) indicates the quantum degeneracy of the ground or excited states of the ion, removed by the magnetic field, and the presence of the *C*—term is due to the difference in Boltzmann populations of the Zeeman sublevels of the ground state of the ion in the external field. At the same time, the *B*—term is associated with the contribution of the “mixing” of wavefunctions owned to the ground or excited states of the ion by the external magnetic field. It is important to note that the *A*— and *C*—terms of the MCD (or FE) characterize the contributions to these magneto-optical effects of magneto-optically active transitions of the type: “singlet → doublet” or “doublet → singlet”, respectively, and can be expressed in terms of matrix elements of these transitions [30,33,34]. In general, the simultaneous existence of contributions of *A*— and *C*—terms is characterized by contributions to the MCD (or FE) of the magneto-active ion of transitions of the type “doublet → doublet”, as well as “doublet → singlet” [30].

The results of low–temperature measurements of the MCD in the studied crystal in the visible region of the spectrum are shown in Figure 6. For comparison, the low-temperature spectrum of the optical density of the KTF crystal is also shown. It is clearly seen that the alternating in sign shape of the MCD signal on the absorption line 2 in Figure 6 is determined by the first derivative of the contour of the same absorption line, or the so-called dispersion shape of the contour of the magneto-optical absorption line, which determines the so-called “diamagnetic” contribution (*A*—term [32,33]) to the MCD of a magnetically active ion. At the same time, the nature of the feature of the MCD spectrum marked as 6″ corresponding to absorption lines 6 and 7 remains unclear and therefore has not been analyzed. It can only be argued that the contribution of line 7 to the MCD is significantly smaller compared to line 6 due to the low absorption intensity associated with it (see Figure 4).

The amplitude of the *A*—term of the MCD is due, on the one hand, to Zeeman splits of doublet sublevels combining in the magneto-optically active transition under consideration, and, on the other hand, its slight increase (~1/*Γ*^2^) correlates with a decrease in the width of the absorption line *Γ* with the temperature. A similar situation is realized, for example, for the 4″ feature of the MCD observed in the vicinity of the absorption line 4, the half-width of which decreases with a decrease in the crystal temperature from 300 to 90 K. It is interesting to note that the spectral dependence of the 4″ MCD feature, measured in the vicinity of this absorption line at *T* = 90 K (Figure 6), looks significantly different from the spectral dependence of the 2″ feature discussed above, and, apparently, can be modeled by a superposition of two opposite (in sign) MCD spectral dependencies corresponding to the “diamagnetic” contributions to the MCD in this region of the spectrum [33] (see inset to Figure 6).

The “diamagnetic” contributions 2″, 4″ and 6″ to the MCD in the KTF crystal considered above occur at magnetically active transitions realized between the states of the Zeeman sublevels of the ground doublet of the lowest multiplet ^7^F_6_ and the Stark sublevels of the doublet and singlet states lying in the excited multiplet ^5^D_4_ (see diagram in Figure 5). Thus, the results of a joint examination of the low-temperature MCD and optical absorption spectra of the KTF crystal made it possible to establish the presence of doublet states (and their position) in the energy spectra of the ^7^F_6_ and ^5^D_4_ multiplets of the Tb^3+^ ion located in the C_4v_ crystal field symmetry.

Further search for doublet states in the spectra of the multiplets of the ground 4*f*^8^ configuration of the Tb^3+^ ion in the KTF crystal considered above can be carried out by comparing its PL and MCPL spectra recorded at the ^5^D_4_ → ^7^F_6_ radiative transition (Figure 7). The results of such a comparison show that the spectrum of the MCPL degree abounds with a large number of features (marked with vertical arrows in Figure 7) with a linear inclined character in the vicinity of the radiation lines. This is especially clearly expressed when considering the features of the 4″ and 6″ spectrum of the MCPL degree, which can be approximated by inclined linear dependencies within the luminescence lines 4″ and 6″, shifted relative to the centers of these lines, as a result of which the magnitude of the effect for them turns out to be asymmetric relative to zero. As for the feature of the 15″ degree of MCPL at a wavelength of 492.5 nm (20,314 cm^−1^), in this case, the MCPL spectrum demonstrates the usual “diamagnetic” dispersion [29,33]. It can be shown that in the approximation of the Gaussian contour of the luminescence band, the expression for the degree of MCPL *P* of the REE ion within the luminescence line (i.e., $\left|\nu -\nu_{0}\right|\approx \Gamma$) can be written as:(4)$$P=\frac{1}{2}\left[\frac{\left(\nu -\nu_{0}\right)}{\Gamma^{2}}g^{\prime }\mu_{B}H+\frac{g\mu_{B}H}{k_{B}T}\right]$$
and represents an inclined linear dependence shifted relative to zero–the center of the radiation line by the value of a temperature-dependent “pedestal” proportional to the amplitude of the so-called “paramagnetic” contribution *C* is a member of the degree of MCPL [30,34]. The first and second terms of dependence (6) determine the contributions of the “diamagnetic” (*A*—term of MCPL) and temperature-dependent “paramagnetic” (*C*—term of MCPL) mechanisms of the magneto-polarized luminescence of the REE ion, respectively: *ν* is the wavenumber (in cm^−1^); *g* and *g*′ are the splitting factors in the magnetic field of the states of the initial and final levels of the radiative transition, and the remaining designations coincide with the designations of the Ref. [30].

Considering the experimental data shown in Figure 7, one can find from them “effective” Zeeman splits of doublet states appearing in radiative transitions of the “doublet–isolated singlet” and “isolated singlet–doublet” types occurring between the Stark sublevels of the Tb^3+^ ion multiplets in the KTF crystal structure. Indeed, since the spectral dependence of the MCPL degree on a complex radiation band at λ~492 nm (20,325 cm^−1^), which also includes the 15′ line, has the usual symmetrical shape for the “diamagnetic” *A*—member of the MCPL (with a change in the effect sign in the center of the line), the feature of the 15′ spectrum of the MCPL degree observed on this line can be unambiguously associated with the radiative transition occurring from the ground Stark sublevel singlet at an energy of 20,560 cm^−1^, located in the lower part of the multiplet ^5^D_4_, to the doublet state of the multiplet ^7^F_6_, lying at an energy of 253 cm^−1^ (see Figure 5).

In this case, the “effective” Zeeman splitting of the doublet state of interest to us in the multiplet under consideration can easily be found using Formula (4) from the doubled product of the magnitude of the angle of inclination of the spectrum of the degree of MCPL — ΔP/Δν by the square of the half-width Γ^2^ of the corresponding radiation line. Since, in the approximation of the Gaussian contour of the luminescence band, the linear dependence on the frequency of the *A*—term of the MCPL degree is valid only within the emission line ($\left|\nu -\nu_{0}\right|\approx \Gamma$) [25,32], it is easy to estimate the half-width Γ (≈5.5 cm^−1^) of the emission line 15′ from experimentally found frequencies of extrema of the linear dependences of the *A*—terms of the MCPL degree for these lines. 

The Zeeman splitting of the doublet state at 253 cm^−1^ in the multiplet ^7^F_6_, found in a similar way and responsible for the occurrence of the *A*—term of the degree of MCPL 15″ on the luminescence line 15′ turns out to be equal to $g\mu_{B}H$ = 1.1 cm^−1^ in the external field *H* = 5 kOe. At the same time, the asymmetry of the spectral dependence of the magnitude of *P* on the radiation lines 4′ and 6′ can be explained by the contribution to the effect of the temperature-dependent “paramagnetic” contribution *C*—degree of MCPL (i.e., by a paramagnetic “pedestal” that shifting the linear dependence of *P* relative to the center of this line). Note that the features of the degree of MCPL on these lines are due to magnetically active transitions occurring between doublet states belonging to states ^5^D_4_ and ^7^F_6_, respectively (see Figure 4).

Thus, the results of a joint examination of the low-temperature spectra of the degree of MCPL and PL of the KTF crystal allow us to establish the presence of three doublet states (and their energies) in the energy spectrum of the ^7^F_6_ multiplet and confirm the presence of two doublet states (and their energies) in the ^5^D_4_ multiplet of Tb^3+^ ions located in the crystal symmetry field C_4v_ (Figure 5) that it agrees well with theoretical predictions [29]. The wavefunctions given in [29] allow us to calculate (see Appendix A) the magnitude of the splitting factor (*g′* = 3.3) in the magnetic field of the doublet state localized at an energy of 253 cm^−1^ in the multiplet ^7^F_6_, which agrees well in order of magnitude with the experimental value found above, equal to 4.7, respectively. As for the ground state of the multiplet ^7^F_6_, the doublet nature of which is confirmed by the data of magneto-optical studies (Figure 5 and Figure 6), then the calculation of its splitting factor in a magnetic field leads to the value *g*′ = 6.94 (see also Appendix A). However, the value of the splitting factor of the ground doublet of the ^7^F_6_ state found above (*g′* = 6.94) turns out to be significantly lower compared to its value (*g′* = 9.54) found from magnetic measurements of the KTF crystal [19] carried out under conditions of saturation of the magnetization of the crystal in a magnetic field of *H* = 80 kOe at *T* = 3 K. As noted by the authors of Ref. [19], a further decrease in the crystal temperature led to a decrease in the magnetization associated with the antiferromagnetic ordering of Tb^3+^ ions, which, in turn, are exposed to the crystal field of C_4v_ symmetry.

Such a discrepancy may be due to the lack of consideration of the contribution to the resulting magnetization of the KTF crystal of the Van-Vleck effect of “mixing” the wavefunctions of the ground doublet with the wavefunction of the nearby singlet lying higher at a distance of 22 cm^−1^ by an external magnetic field *H* (see Figure 5). For example, the calculation of the temperature-independent Van-Vleck additive to the resulting magnetization of the studied crystal, performed in the Appendix A, shows that at low temperatures, its value is comparable to the average magnetic moment of the RE-ion due to the different Boltzmann population of the sublevels of the ground doublet state of the ^7^F_6_ multiplet.

## 4. Conclusions

Thus, the results of the above technological and experimental studies allowed:Successfully growing the cubic crystals KTF characterized by high optical quality and bright visible PL;Performing precision studies of the absorption, PL, MCD, and MCPL spectra at temperatures *T* = 90 and 300 K in grown crystals in the absorption (emission) band associated with the 4*f*-4*f* forbidden transition ^5^D_4_ → ^7^F_6_, localized in the spectral range 480–500 nm (20,800–20,000 cm^−1^);Constructing the optical transitions scheme in the absorption and PL spectra in the KTF crystal and determining the energies of most of the Stark sublevels (singlets and doublets) of the ground ^7^F_6_ and excited ^5^D_4_ multiplets of the Tb^3+^ ion in the crystal environment of C_4v_ symmetry;From a joint examination of the low-temperature spectra of the degree of MCPL, MCD, PL and optical absorption of the KTF crystal, establishing the presence of three doublet states (and their energies) in the energy spectrum of the ^7^F_6_ multiplet and confirming the presence of two doublet states in the ^5^D_4_ multiplet of Tb^3+^ ions located in a crystal field of tetragonal symmetry confirmed by theoretical predictions;Based on the use of the wave functions of the Stark sublevels of multiplets split by a crystal field of tetragonal symmetry and combining both in the optical radiative ^5^D_4_ → ^7^F_6_ and in the absorption ^7^F_6_ → ^5^D_4_ transitions in the Tb^3+^ ion, give a realistic explanation of the magnetic and magneto-optical properties of the KTF crystal.

## Figures and Tables

**Figure 1 materials-15-07999-f001:**
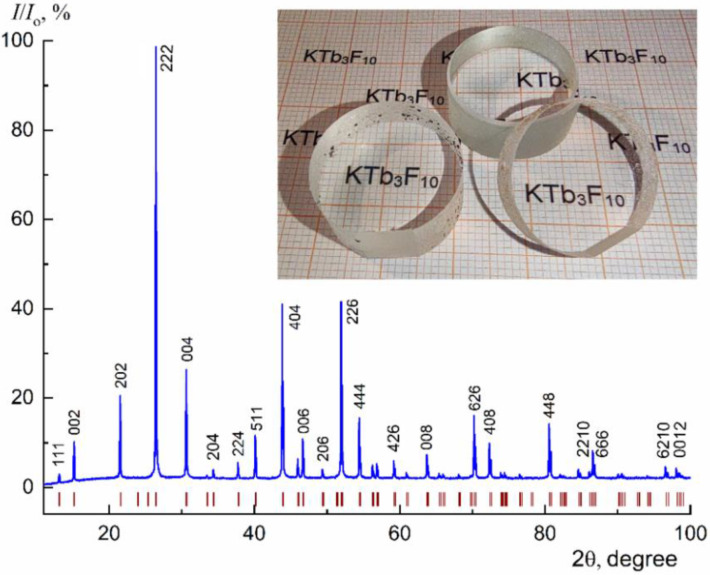
XRD pattern of the KTF crystal. The positions of the Bragg reflections peak within sp. gr. *Fm*–3*m* (the lattice parameter *a* = 11.6732(2) Å) are indicated. Optical elements fabricated from the grown crystals are demonstrated in the insert.

**Figure 2 materials-15-07999-f002:**
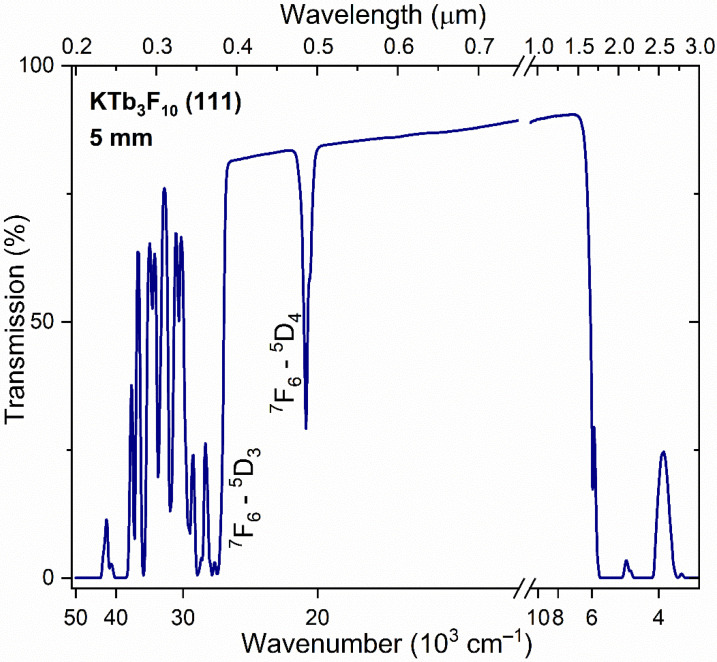
RT transmission spectrum of the KTF crystal.

**Figure 3 materials-15-07999-f003:**
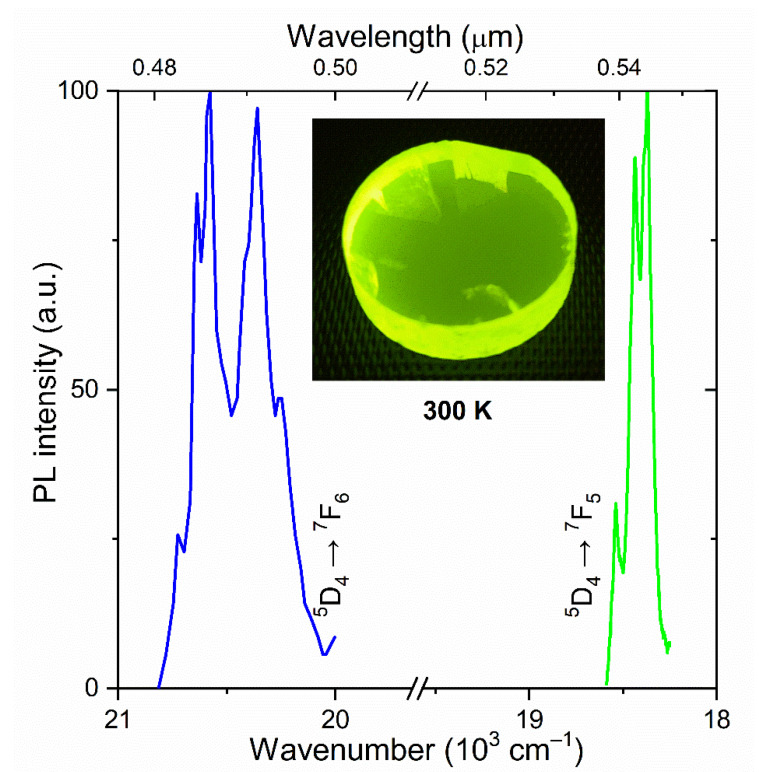
The normalized PL spectra of a KTF crystal at RT. Insert: the visible PL of the investigated crystals under UV excitation.

**Figure 4 materials-15-07999-f004:**
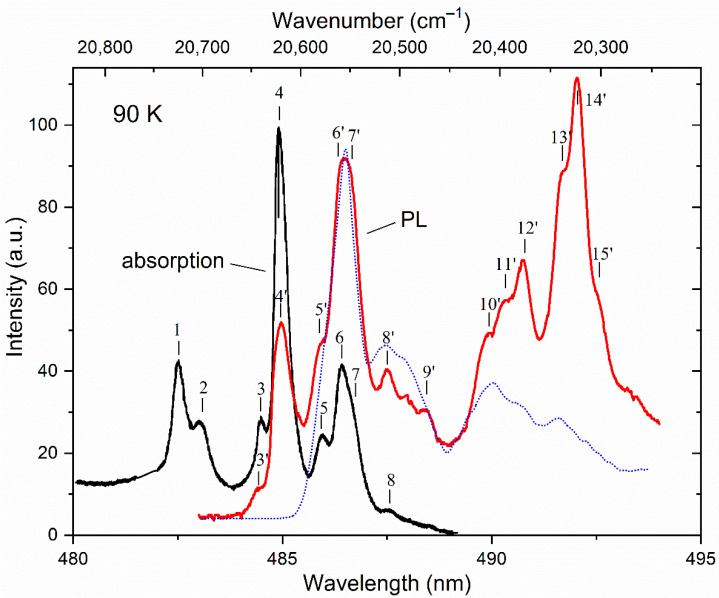
Low-temperature absorption (black) and PL (red) spectra of a KTF crystal under Hg lamp excitation. The PL data measured at *T* = 3 K [19] are shown as a dotted line for comparison. Peaks in the spectra are indicated by numbers.

**Figure 5 materials-15-07999-f005:**
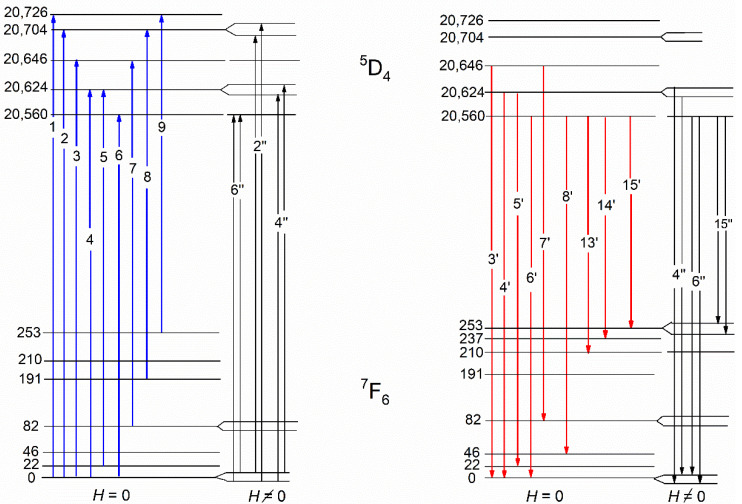
The scheme of Tb^3+^ optical transitions in a KTF crystal, constructed according to the absorption (blue arrows) and PL (red arrows) spectra at *T* = 90 K. The diagrams of magnetically–optically active transitions in Tb^3+^ ions, realized in the magnetic field *H* in the low-temperature spectra of MCD and MCPL, are shown on the right parts.

**Figure 6 materials-15-07999-f006:**
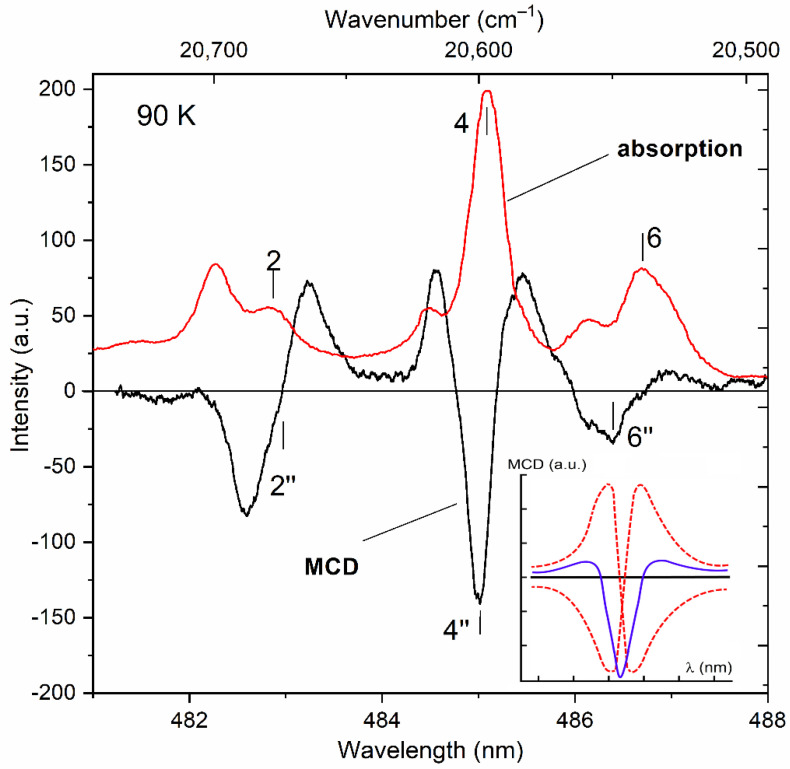
Spectral dependences of the MCD (red solid) and optical absorption (black line) of the KTF crystal at *T* = 90 K. The MCD spectrum is recorded in a magnetic field *H* = 5 kOe. The characteristic features of the MCD and absorption spectra are indicated in accordance with the transition scheme (Figure 5). The insert shows the result of the superposition of two opposite (in sign) spectral dependences of the MCD corresponding to the “diamagnetic” contributions to the MCD in the studied spectral region.

**Figure 7 materials-15-07999-f007:**
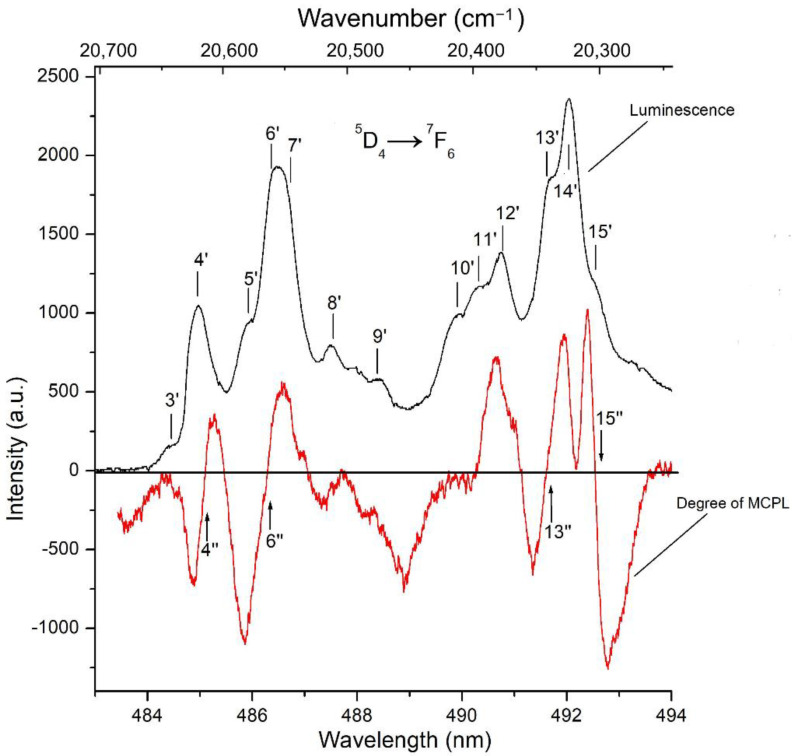
The spectra of the degree of MCPL (red solid) and PL (black line) of the KTF crystal at a temperature of *T* = 90 K in the visible spectral region. The MCPL spectrum is recorded in a magnetic field *H* = 5 kOe. The characteristic features of the spectra are numbered in accordance with the transitions scheme shown in Figure 5.

**Table 1 materials-15-07999-t001:** The energy *E* (cm^−1^) of optical transitions between the ground ^7^F_6_ and excited ^5^D_4_ multiplets of the Tb^3+^ ion observed in the absorption and PL spectra of the KTF crystal at *T* = 90 K.

Absorption		Luminescence	
Transition Label	*E*	Transition Label	*E*
1	20,726	3′	20,646
2	20,704	4′	20,624
3	20,646	5′	20,602
4	20,624	6′	20,560
5	20,602	7′	20,564
6	20,560	8′	20,513
7	20,564	9′	20,478
8	20,513	10′	20,409
9	20,473	11′	20,393
		12′	20,371
		13′	20,350
		14′	20,323
		15′	20,307

## Data Availability

Not applicable.

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
