# Peer review of "Tb3+ Ion Optical and Magneto-Optical Properties in the Cubic Crystals KTb3F10"

_materials, 2022, doi:10.3390/ma15227999_

Round 1

Reviewer 1 Report

The Authors have presented the work properly and the manuscript is well-written with an adequate explanation of the results.

I accept this work to be published in the present version. 

- The authors have used the wavefunctions of the Stark sublevels of multiplets split obtained from the tetragonal crystal field and combined their study with the optical transition. They also provide additional information regarding the magnetic and magneto-optical properties observed in the KTb3F10 crystal.

- The topic is quite not so original and there are few kinds of literature dealing with similar problems. However, their main aim is to grow optical-quality 50 KTF crystals from the melt and the analysis of experimental data from optical and magneto-optical measurements of these single crystals in the visible spectral range.

- According to me, they did their experiments and explained the data quite nicely. However, if possible can put traces in the literature survey.  

- The conclusions are okay.

- No comments regarding the table. However, for a good representation, the authors can give grid lines on the right side of the figure.

Author Response

Reviewer 1

The Authors have presented the work properly and the manuscript is well-written with an adequate explanation of the results.

I accept this work to be published in the present version. 

- The authors have used the wavefunctions of the Stark sublevels of multiplets split obtained from the tetragonal crystal field and combined their study with the optical transition. They also provide additional information regarding the magnetic and magneto-optical properties observed in the KTb3F10 crystal.

- The topic is quite not so original and there are few kinds of literature dealing with similar problems. However, their main aim is to grow optical-quality 50 KTF crystals from the melt and the analysis of experimental data from optical and magneto-optical measurements of these single crystals in the visible spectral range.

- According to me, they did their experiments and explained the data quite nicely. However, if possible can put traces in the literature survey.  

- The conclusions are okay.

- No comments regarding the table. However, for a good representation, the authors can give grid lines on the right side of the figure.

Authors’ response 

We would like to express our gratitude to the reviewer for checking the manuscript and making minor comments that will help make it clearer for readers and more perfect.

According to the reviewer’s request, we expanded the scope and content of the manuscript's Introduction.

As for giving the grid lines, we understand them as a scale grid. Are we right? But adding this grid makes figures difficult to read due to too many details in the figure. Moreover, some figures have two Y-axes.

Reviewer 2 Report

This manuscript reported optical and magneto-optical properties in the KTb3F10. Optical properties have been previously reported in the ref.18, but magneto-optical properties have been well investigated in this work, and, the properties were well discussed. The reviewer recommends minor revisions as follows.

1.       Please introduce the previous study on ref.18 (J. Lumin. 2020, 227, 117523) in the introduction section in order to show scientific advance in this work.

2.       It is a little unclear why the authors chose KTF for the detailed study of magneto-optical properties.

Author Response

Reviewer 2

This manuscript reported optical and magneto-optical properties in the KTb3F10. Optical properties have been previously reported in the ref.18, but magneto-optical properties have been well investigated in this work, and, the properties were well discussed. The reviewer recommends minor revisions as follows.

  1. Please introduce the previous study on ref.18 (J. Lumin. 2020, 227, 117523) in the introduction section in order to show scientific advance in this work.
  2. It is a little unclear why the authors chose KTF for the detailed study of magneto-optical properties.

Authors’ response 

According to the reviewer’s comment, we provide a more detailed description of the previous research results from Ref. 19 (J. Lumin. 2020, 227, 117523) in the Introduction section:

“In our opinion, the use of magneto-optical research methods has significantly expanded the capabilities of traditional optical methods previously used by Pues et al. [19] to the REE compound under consideration, which made it possible to establish both the presence of doubly degenerate states (doublets) and their energies in the spectra of 5D4 and 7F6 multiplets of Tb3+ ions localized in the tetragonal crystal field of C4v symmetry of KTF crystals.”

The reason why we choose KTF as the object of magneto-optical research is that we hope to realize an effect of considerable change of the circularly-polarized luminescence induced by the external magnetic field at 90 K, which may have interest in practical applications. In this case, it is worth noting that a similar effect at T = 90 K has been already detected in the terbium-yttrium aluminum garnet, and its value reaches 45% in the green luminescence band at magnetic field of 4.5 kOe. In addition, a large number of our data and data from Ref.19 enable us to construct the Stark splitting diagram of 7F6 and 5D4 multiples of Tb3+ ion in KTF crystal structure.

Reviewer 3 Report

The article presents the growth of KTb3F10 (KTF) crystals and their optical and magneto-optical properties in the visible spectral range (480–500nm). Lately, KTF crystals have been manufactured as high-power laser isolators. The research design is appropriate, and the methods for crystal growth and their optical and magneto-optical characterization are convincingly presented. The results on absorption and photoluminescence, magneto-optical, and circular magnetic polarization of luminescence at T = 90 and 300 K are clearly presented, and the results support the conclusions.   

However, the introduction shapes from a small text and does not emphasize the specific problem (motivation), knowledge gaps, or the importance and added value of the research topic. Wide band gap fluoride crystals doped with Tb3+ were used for different spectral region applications over many years, e.g., https://doi.org/10.1016/S0030-4018(98)00367-8,  https://doi.org/10.1007/s00340-022-07759-1,  https://doi.org/10.1016/j.pquantelec.2022.100411). The authors should compare the optical properties (e.g., band gap) of the KTF crystal with the other wide band gap fluoride crystals doped with Tb3+ matrixes in the literature and justify why the present KTF crystal is superior compared with other crystals in previous publications. A common problem of wide band gap fluoride dielectric crystals of trivalent rare earth ions under high-power laser irradiation is the formation of permanent or transient color centers, leading to degradation of the crystal's optical properties (compactness). This is an application-oriented bottlenecking effect that must be discussed in the article. The authors should refer to a KTF crystal-oriented application. The Stark splitting differences correspond to ~660 GHz. Do the authors aim for applications in this spectral region?  

Minor improvement points should be corrected, such as

Page 2, line 65 "range of angles 2θ from 10 to 120°."; But in Fig.1, up to 100° is shown.

Page 5, line 183 "Absorption and FL spectra of KTF.."; "FL" should be defined or replaced by PL.

After improvement according to the above lines, the article is suitable for publication in the MATERIALS  journal.

Author Response

Reviewer 3

The article presents the growth of KTb3F10 (KTF) crystals and their optical and magneto-optical properties in the visible spectral range (480–500nm). Lately, KTF crystals have been manufactured as high-power laser isolators. The research design is appropriate, and the methods for crystal growth and their optical and magneto-optical characterization are convincingly presented. The results on absorption and photoluminescence, magneto-optical, and circular magnetic polarization of luminescence at T = 90 and 300 K are clearly presented, and the results support the conclusions.   

However, the introduction shapes from a small text and does not emphasize the specific problem (motivation), knowledge gaps, or the importance and added value of the research topic. Wide band gap fluoride crystals doped with Tb3+ were used for different spectral region applications over many years, e.g., https://doi.org/10.1016/S0030-4018(98)00367-8,  https://doi.org/10.1007/s00340-022-07759-1,  https://doi.org/10.1016/j.pquantelec.2022.100411). The authors should compare the optical properties (e.g., band gap) of the KTF crystal with the other wide band gap fluoride crystals doped with Tb3+ matrixes in the literature and justify why the present KTF crystal is superior compared with other crystals in previous publications. A common problem of wide band gap fluoride dielectric crystals of trivalent rare earth ions under high-power laser irradiation is the formation of permanent or transient color centers, leading to degradation of the crystal's optical properties (compactness). This is an application-oriented bottlenecking effect that must be discussed in the article. The authors should refer to a KTF crystal-oriented application. The Stark splitting differences correspond to ~660 GHz. Do the authors aim for applications in this spectral region?  

Minor improvement points should be corrected, such as

Page 2, line 65 "range of angles 2θ from 10 to 120°."; But in Fig.1, up to 100° is shown.

Page 5, line 183 "Absorption and FL spectra of KTF.."; "FL" should be defined or replaced by PL.

Authors’ response

According to the reviewer’s comment:

«However, the introduction shapes from a small text and does not emphasize the specific problem (motivation), knowledge gaps, or the importance and added value of the research topic»

we provide a more detailed description

“In our opinion, the use of magneto-optical research methods has significantly expanded the capabilities of traditional optical methods previously used by Pues et al. [19] to the REE compound under consideration, which made it possible to establish both the presence of doubly degenerate states (doublets) and their energies in the spectra of 5D4 and 7F6 multiplets of Tb3+ ions localized in the tetragonal crystal field of C4v symmetry of KTF crystals. It is this experimental discovery of doublet states in the spectra of 5D4 and 7F6 multiplets that made it possible to give a consistent explanation for the appearance of features in the magnetic and magneto-optical properties of KTF crystals.”

As for the reviewer’s comment: «Wide band gap fluoride crystals doped with Tb3+ were used for different spectral region applications over many years, e.g., https://doi.org/10.1016/S0030-4018(98)00367-8,https://doi.org/10.1007/s00340-022-07759-1,  https://doi.org/10.1016/j.pquantelec.2022.100411). The authors should compare the optical properties (e.g., band gap) of the KTF crystal with the other wide band gap fluoride crystals doped with Tb3+ matrixes in the literature and justify why the present KTF crystal is superior compared with other crystals in previous publications»

These works are well known to us. The prospects of terbium crystal research has aroused great interest. However, our main task is not to compare different crystals with terbium, but to study the Stark structure and magneto-optical properties of KTF in detail based on a large number of experimental data obtained by us and the authors in Ref.19. It is worth noting that this crystal seems to be very interesting and relevant for magneto-optical applications, and its spectroscopic properties have not been completely determined.

Next comment: «A common problem of wide band gap fluoride dielectric crystals of trivalent rare earth ions under high-power laser irradiation is the formation of permanent or transient color centers, leading to degradation of the crystal's optical properties (compactness). This is an application-oriented bottlenecking effect that must be discussed in the article»

In fact, it is not the goal of this work to discuss these problems of KTF crystals, moreover, we do not work in the UV spectral region.

Minor improvements:

Page 2, line 65 "range of angles 2θ from 10 to 120°."; But in Fig.1, up to 100° is shown -

Corrected.

Page 5, line 183 "Absorption and FL spectra of KTF.."; "FL" should be defined or replaced by PL

Corrected.